# Expert or not? assessing data quality in offline reinforcement learning

## Abstract

Offline reinforcement learning (RL) learns exclusively from static datasets, without further interaction with the environment. In practice, such datasets vary widely in quality, often mixing expert, suboptimal, and even random trajectories. The choice of algorithm therefore depends on dataset fidelity: behavior cloning can suffice on high-quality data, whereas mixed- or low-quality data typically benefits from offline RL methods that stitch useful behavior across trajectories. Yet in the wild it is difficult to assess dataset quality a priori because the data's provenance and skill composition are unknown. We address the problem of estimating offline dataset quality *without* training an agent. We study a spectrum of proxies—from simple cumulative rewards to learned value-based estimators—and introduce the *Bellman–Wasserstein distance (BWD)*, a value-aware optimal-transport score that measures how dissimilar a dataset's behavioral policy is from a random reference policy. BWD is computed from a behavioral critic and a state-conditional OT formulation, requiring no environment interaction or full policy optimization. Across D4RL MuJoCo tasks, BWD strongly correlates with an *oracle* performance score that aggregates multiple offline RL algorithms, enabling efficient prediction of how well standard agents will perform on a given dataset. Beyond prediction, integrating BWD as a regularizer during policy optimization explicitly pushes the learned policy away from random behavior and improves returns. These results indicate that value-aware, distributional signals such as BWD are practical tools for triaging offline RL datasets and policy optimization.

## 1 Introduction

Deep reinforcement learning has achieved notable advancements in diverse domains such as robotics (Mnih et al., 2015; Zhao et al., 2020), complex game environments (Vinyals et al., 2019), and dialog systems (Ouyang et al., 2022). In recent years, offline RL has emerged as a significant area of focus within the RL research community. The offline RL paradigm distinguishes itself from standard RL by operating exclusively on a pre-existing dataset of experiences, without any interaction with the environment (Levine et al., 2020). The practical utility of offline settings is evident in domains where online interactions are inherently risky, time-consuming, or ethically problematic, such as in robotics, autonomous vehicle development, and medical treatment.

The choice of an appropriate algorithm for offline RL depends heavily on the quality of the available data. For instance, when working with datasets that are mostly made up of high-quality expert demonstrations, standard behavior cloning (BC) can be an efficient and relatively simple solution (Kumar et al., 2022). Conversely, when datasets exhibit a wider spectrum of expertise levels, it becomes imperative to employ more sophisticated offline RL algorithms capable of effectively stitching together various trajectories (Wu et al., 2019; Brandfonbrener et al., 2021; Kostrikov et al., 2021; Fujimoto & Gu, 2021). Although complex offline RL algorithms are more efficient with suboptimal data, almost all of them incorporate a behavior cloning component during training Levine et al. (2020). Depending on the data, various levels of reliance on the behavior cloning component need to be set, see Figure 1.

In practice, collecting datasets composed solely of expert-level interactions is often problematic. As a result, the provided datasets frequently comprise a mixture of expert-level demonstrations and suboptimal or even random trajectories. To simulate real-world conditions and rigorously evaluate algorithmic performance across varied settings, data in prominent offline RL benchmarks, such as

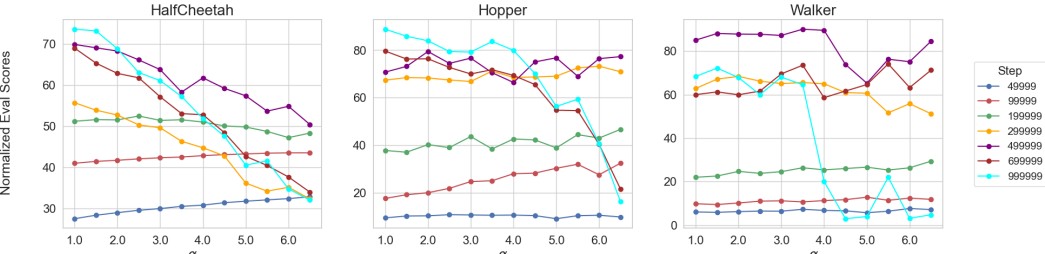

Figure 1: The impact of the hyperparameter $\alpha$ for TD3+BC Fujimoto & Gu (2021), which balances: RL and BC. 12 different values of the $\alpha$ parameter were tested. The average normalized scores are reported over the different training steps and three random seeds. As the value of $\alpha$ increases, the performance on high-quality datasets decreases or not changing, while the performance on lower-quality datasets improves.

D4RL (specifically, `Mujoco` environments Fu et al. (2020)), are systematically categorized based on the quality of the experiences, often labeled as random, medium, expert, and so on. In the wild, we need to train various agents to estimate the overall data quality.

These observations motivate a research question: *How can we effectively measure data quality without the computationally expensive process of training an agent?* To systematically address this overarching question, we decompose it into three specific sub-questions:

- Is there a discernible relationship between the cumulative rewards and Q-values derivable from the dataset and the ultimate performance achieved by an agent trained on this data?

- If we posit that a random policy represents the lowest performance, can the deviation of the dataset's policy from this random baseline serve as a reliable measure of data quality?

- Can adopting a distributional perspective on the data provide valuable insights for estimating its intrinsic quality?

We investigate the concept of difference between a random policy and the behavioral policy implicit in the dataset. Building upon this, we introduce a novel metric, termed the *Bellman-Wasserstein Distance* (BWD), which leverages the $Q$-function to quantify the dissimilarity between the random policy and the behavioral policy. We demonstrate that employing this BWD as an additional regularization term during policy optimization can lead to improvements in policy performance.

*We make the following contributions:*

- **Bellman–Wasserstein distance (BWD).** We introduce a value-aware optimal-transport score for offline RL that measures the dissimilarity between the dataset's behavioral policy and a random reference policy. Concretely, BWD couples actions $(a, a')$ *state-conditionally* and uses the cost $c((s, a), (s, a')) = Q_\beta(s, a') - \|a' - a\|_2^2$, optimized via the entropically regularized dual (Eq. (14)–(16)). This requires only a behavioral critic $Q_\beta$ learned from the static dataset (Sec. 4).

- **Predictive power for data quality.** Across D4RL MuJoCo tasks (HalfCheetah, Hopper, Walker2d), BWD *increases with dataset quality* and achieves the strongest correlation with an oracle score that aggregates multiple offline RL algorithms, outperforming reward-, $Q_\beta$-, advantage-, and PD-based baselines (Table 1; Fig. 2; Sec. 5).

- **Sample and compute efficiency.** Computing BWD uses a small subset of transitions and converges with $\sim$10k updates on a single consumer GPU, making it far cheaper than training a full policy for evaluation ("Run time", Sec. 5).

- **Regularization for policy learning.** We show that adding BWD as a regularizer to IQL improves normalized returns on standard locomotion tasks, especially on Walker2d and HalfCheetah, validating that explicitly pushing the learned policy away from random behavior is beneficial in offline settings (Table 2; Fig. 3; Sec. 6).

- **Benchmarking resources.** To probe finer-grained quality effects, we curate extended D4RL-style splits with seven quality levels per environment and will release code and scripts for reproducibility (Sec. 5).

## 2 PRELIMINARIES

### 2.1 REINFORCEMENT LEARNING

Reinforcement learning is an established framework for decision making in Markov decision processes (MDPs). An MDP is defined by the state space $S$, the action space $A$, and their corresponding distributions $\mathcal{S}$ and $\mathcal{A}$. It also includes the transition distribution $T(s' \mid s, a)$, the reward function $r(s, a)$, and the discount factor $\gamma$. The goal in RL is to find a policy $\pi(a|s)$ that maximizes the expected cumulative reward over time $t$: $\mathbb{E}_\pi \left[ \sum_{t=0}^{H} \gamma^t r(s_t, a_t) \right]$, where $H$ can be infinite.

The action value function $Q^\pi$ is a crucial component in the reinforcement learning framework. It provides an estimate of the expected cumulative reward that can be obtained by following a given policy $\pi$. Mathematically, this can be written as

$$Q^\pi(s, a) = r(s, a) + \gamma \mathbb{E}_{s' \sim T(s,a), a' \sim \pi(s')} \left[ Q^\pi(s', a') \right], \tag{1}$$

where $s'$ and $a'$ are the next state and the next action, respectively. This recursive formulation allows the action value function $Q^\pi$ to be estimated by iteratively updating its estimate until convergence is reached. In practice, when we the parameters $Q^\pi$ by e.g. neural networks with parameters $\phi$ are updated by minimizing the mean squared Bellman error over the experience-replay dataset $(s, a, s', r) \leftarrow \mathcal{D}$:

$$\min_{Q^\pi_\phi} \mathbb{E}_{(s,a,s') \sim \mathcal{D}} \left[ \left( r(s, a) + \gamma \mathbb{E}_{a' \sim \pi_\theta(s')} \left[ Q^\pi_\phi(s', a') \right] - Q^\pi_\phi(s, a) \right)^2 \right]. \tag{2}$$

By correctly estimating the action value function, a near-optimal policy can be recovered. The Deterministic Policy Gradient (DPG) Silver et al. (2014) method can be used for continuous control tasks. This method simply updates the actor in the direction of the gradient of the action-value function with respect to the action. When combined with deep neural networks, this algorithm is called Deep Deterministic Policy Gradient (DDPG) Lillicrap et al. (2015). Its improved version named Twin Delayed Deep Deterministic Policy Gradient (TD3) Fujimoto et al. (2018) finds a rich application in continuous control.

### 2.2 OFFLINE REINFORCEMENT LEARNING

Offline RL and behavior cloning (BC) approaches provide a fully data-driven approach that does not require interactions with the environment Levine et al. (2020). The simplest BC objective is a squared error $(\pi(s) - a)^2$ between policy $\pi(s)$ and actions $a$ provided by the expert policy dataset $\mathcal{D}$ Pomerleau (1991). The policy optimization problem for the behavior cloning can be defined as:

$$\min_\pi \mathbb{E}_{(s,a) \sim \mathcal{D}} [(\pi(s) - a)^2]. \tag{3}$$

However, datasets often contain both expert and potentially irrelevant task behaviors, and BC algorithms can perform poorly. To avoid this problem, offline RL algorithms combine useful behavior segments spread over multiple suboptimal trajectories Kumar et al.. Recently, a computationally efficient approach to offline RL algorithms called TD3+BC has been proposed Fujimoto & Gu (2021). This algorithm by design combines offline RL and BC components Eq.3 by multiplying the behavior cloning term by the parameter $\lambda > 0$:

$$\max_\pi \mathbb{E}_{(s,a) \sim \mathcal{D}} [\lambda Q(s, \pi(s)) - (\pi(s) - a)^2]. \tag{4}$$

Although this formulation is simple, it has performed admirably on a variety of tasks. Another family of methods *without off-policy evaluation* method (Brandfonbrener et al., 2021). These approaches approximate the behavioral $Q^\beta$ function using the provided with the dataset of experience, so these method learn a $Q$ function using the SARSA algorithm:

$$\min_{Q^\pi_\phi} \mathbb{E}_{(s,a,r,s') \sim \mathcal{D}} \left[ \left( r + \gamma \mathbb{E}_{a' \sim \mathcal{D}(s')} \left[ Q^\pi_\phi(s', a') \right] - Q^\pi_\phi(s, a) \right)^2 \right]. \tag{5}$$

After obtaining a $Q^\beta$ function, it using to extract the corresponding policy $\pi$ by weighting actions with the advantage function $A^\beta(s, a)$ (Sutton & Barto, 2018). More improved methods like IQL approximate the policy improvement by treating the state value function $V^\beta(s)$ as a random variable and taking a state-conditional upper expectile to estimate the value of the best actions.

### 2.3 WASSERSTEIN DISTANCE

To understand the nature of Wasserstein distance, we give a brief description of optimal transport problems. The Monge problem Villani (2008) represents the classical formulation of OT. It seeks an optimal transport map $T$ that transforms a probability distribution $\mu$ (source) into another probability distribution $\nu$ (target), while minimizing a total transportation cost defined by a cost function $c(x, y)$. The map $T$ must satisfy $T_{\#}\mu = \nu$, where $T_{\#}\mu$ is the push-forward measure of $\mu$ by $T$. Formally, Monge's problem is:

$$\min_{T\sharp\mu=\nu} \mathbb{E}_{x\sim\mu}[c(x, T(x))]. \tag{6}$$

The cost function $c(x, y)$ measure how hard it is to move a mass piece between points $x \in \mathcal{X}$ and $y \in \mathcal{Y}$ from distributions $\mu$ and $\nu$ correspondingly. That is, an OT map $T$ shows how to optimally move the mass of $\mu$ to $\nu$, i.e., with the minimal effort.

Kantorovich (Kantorovitch, 1958) introduced a relaxed formulation of OT that allows for mass splitting and ensures a solution always exists. It is important because for certain values of $\mu$ and $\nu$ there may be no map $T$ that satisfies $T_{\#}\mu = \nu$. The Kantorovich OT problem can be written as

$$\min_{\gamma\in\Pi(\mu,\nu)} \mathbb{E}_{x,y\sim\gamma}[c(x, y)]. \tag{7}$$

In this case, the minimum is obtained over the transport plans $\gamma$, which refers to the couplings with the respective marginals being $\mu$ and $\nu$. The optimal $\gamma^*$ belonging to $\Pi(\mu, \nu)$ is referred to as the *optimal transport plan*. If entropy or $L^p$ regularizations are used, then the problem can then be defined as a regularized OT problem Seguy et al. (2017). For cost functions $c(x, y) = \|x - y\|_2$ and $c(x, y) = \frac{1}{2}\|x - y\|_2^2$, the OT cost is called the Wasserstein-1 and the (square of) Wasserstein-2 distance, respectively, see (Villani, 2008, 1)(Santambrogio, 2015, 1, 2).

The application of OT methods and Wasserstein distances has seen a surge in machine learning, particularly within generative modeling. A significant line of research focuses on employing neural networks to compute or approximate optimal transport maps and distances Makkuva et al. (2019); Korotin et al. (2021a); Gulrajani et al. (2017); Korotin et al. (2022c); Asadulaev et al. (2022). In our paper we use these neural network based methods methods. This allows computing a continuously differentiable distance that can be implemented in the continuous state and action space.

## 3 OFFLINE DATA QUALITY ESTIMATION

Offline RL setting, there is no interaction with the environment (Levine et al., 2020). Therefore, estimating the solvability of the task shifts to estimating the quality of the data. Recent results have shown that if assumptions can be made about the quality of the data, no algorithm can outperform standard behavioral cloning if your data consists only of perfect data; in such scenarios, standard behavioral cloning is a relatively simple and correct choice (Kumar et al., 2022). However, the BC approach fails when the data is noisy or suboptimal. For noisy data, or data obtained from multiple policies that perform well on different parts of the environment, offline RL can achieve significantly better results (Levine et al., 2020; Fujimoto et al., 2018; Kumar et al., 2022).

To understand the quality of the provided data, we can train an agent and compute the cumulative reward. However, there are many different algorithms and the scores can vary significantly, See Table1 by Kostrikov et al. (2021) for the example. Therefore, to estimate the quality of the data, we need to train a bunch of different algorithms on the given problem. Yet, brute forcing the problem with different algorithms can be resource inefficient (Furuta et al., 2021).

To avoid brute-force way for data quality estimatiom in this section explores several established metrics for evaluating the quality of offline reinforcement learning datasets without necessitating the full training of an RL agent. These methods provide insights into the dataset's potential by analyzing inherent properties of the collected trajectories and associated value functions.

**Cumulative Reward**. Using the dataset provided by the expert, the simplest way to estimate data quality is to calculate the average reward obtained from the dataset

$$\mathbb{E}_{s,a,r\sim\mathcal{D}}[r(s, a)]. \tag{8}$$

However, the issue is that reward values can vary significantly in different environments (Furuta et al., 2021). When using the average reward value, one may obtain a higher value even if the rewards are

sparse (Rengarajan et al., 2022). If one of the states $s$ yields a high reward, it can inflate the average value. Therefore, more sophisticated methods are needed to estimate the data quality.

**Q-function**. The second method involves the $Q$ function, according to the $\beta$ policy provided in the dataset. Although the optimal $Q^\beta(s, a)$ value is typically unknown, we can use a critic function $Q^\beta_\theta : \mathbb{R}^S \times \mathbb{R}^A \to \mathbb{R}$, approximated by a neural network trained on the available expert data. The SARSA method (Sutton & Barto, 2018) allows for the behavioral $Q$-function to be reconstructed from the data:

$$Q^\beta_\theta(s, a) = [(r(s, a) + \gamma \mathbb{E}_{s' \sim \mathcal{D}(s,a), a' \sim \beta(s')}[Q^\beta_\theta(s', a')]. \tag{9}$$

Further, the average of the obtained $Q^\beta$ over the data can also be used as an estimation.

$$\mathbb{E}_{s,a \sim \mathcal{D}}[Q^\beta_\theta(s, a)]. \tag{10}$$

Training the Q-function with a discount factor $\gamma$ close to 1 can mitigate issues associated with simpler metrics like average reward, particularly in sparse reward environments.

**Advantage function**. The advantage function, $A^\beta(s, a)$, quantifies the relative merit of taking action $a$ in state $s$ compared to the average action under the behavioral policy $\beta$. It is a widely utilized concept in reinforcement learning for assessing action quality (Sutton & Barto, 2018; Schulman et al., 2017; Lillicrap et al., 2015; Wang et al., 2020). The advantage can be estimated using the previously learned behavioral Q-function, $Q^\beta_\theta$:

$$A^\beta(s, a) = Q^\beta_\theta(s, a) - \mathbb{E}_{a \sim \beta(s)}[Q^\beta_\theta(s, a)]. \tag{11}$$

Here, by definition, $\mathbb{E}_{a \sim \mathcal{D}}[Q^\beta_\theta(s, a)] = V^\beta(s)$. The expectation of the advantage function over the entire dataset

$$\mathbb{E}_{(s,a) \sim \mathcal{D}}[A^\beta(s, a)], \tag{12}$$

provides a measure of how much, on average, the actions in the dataset outperform the mean behavioral policy's actions. This is insightful in offline RL, where datasets may originate from multiple experts of varying quality. The advantage function thus helps to discern how many actions within the dataset are demonstrably superior to the average behavior observed for their respective states.

**Performance-Difference**. As established by Kakade & Langford (2002), provides a formal means to compare the performance of two policies, $\pi$ and $\beta$:

$$J(\pi) - J(\beta) = \frac{1}{1 - \gamma} \mathbb{E}_{s \sim d^\pi} \left[ A^\beta(s, \pi(s)) \right], \tag{13}$$

where $d^\pi$ is the discounted state visitation distribution induced by policy $\pi$, and $A^\beta(s, a) = Q^\beta(s, a) - V^\beta(s)$ is the advantage function of policy $\beta$. Using it let's consider $\beta$ to be an policy learned using the provided data, and $\pi$ to denote any other policy. Given any policy $\pi$, we define its performance as $J(\pi) \overset{def}{=} \underset{\tau \sim \pi}{\mathbb{E}} [\sum_{t=0}^\infty \gamma^t R(s_t, a_t, s_{t+1})]$, where $\tau \sim \pi$ denotes the random trajectories obtained by following the policy $\pi$, and $(s_0 \sim S, a_t \sim \pi(s_t), s_{t+1} \sim P(\cdot \mid s_t, a_t))$. However, *it is unclear, which policy should be used as $\pi$?* We answer this in Section 5.

## 4 BELLMAN-WASSERSTEIN DISTANCE

In the context of offline RL, a random policy often serves as a baseline representing the least effective behavior. Our idea is to quantify how significantly the behavior policy deviates from a random baseline. We introduce the Bellman-Wasserstein Distance, a novel metric that leverages optimal transport to measure this deviation in a value-aware manner. The BWD evaluates the dissimilarity between the empirical state-action distribution $P_\beta$ derived from $\mathcal{D}$ (where $(s, a) \sim \mathcal{D}$) and a reference distribution $P_{\text{rand}}$. The latter is composed of pairs $(s, a')$, where states $s$ are drawn from the marginal state distribution in $\mathcal{D}$ (denoted $s \sim \mathcal{D}_S$) and actions $a'$ are sampled from the random policy, $a' \sim \pi_{\text{rand}}(\cdot|s)$.

While OT commonly employs costs based on Euclidean distances (Villani, 2008), We propose to consider a Q function as cost: For a pair $(s, a) \in \text{supp}(P_\beta)$ and $(s, a') \in \text{supp}(P_{\text{rand}})$ (sharing the same state $s$), we define the cost as

$$c((s, a), (s, a')) = Q^\beta_\theta(s, a') - \|a' - a\|^2_2, \tag{14}$$

where $Q_\theta^\beta(s, a')$ is the Q-value of the random action $a'$ in state $s$, estimated by a Q-function $Q_\theta^\beta$ that models the expected return of the behavior policy $\beta$. The term $\|a' - a\|_2^2$ is the squared Euclidean distance between actions $a'$ and $a$. A smaller BWD indicates that $P_\beta$ can be optimally coupled with $P_{\text{rand}}$ at a low total cost. This occurs when, on average, random actions $a'$ yield low values under $Q_\theta^\beta$ (small $Q_\theta^\beta(s, a')$) and behavioral actions $a$ are substantially different from these random actions (large $\|a' - a\|_2^2$).

Computationally, the BWD is determined via the dual formulation of the entropically regularized Kantorovich problem (Kantorovitch, 1958; Seguy et al., 2017). This approach is often more tractable than solving the primal problem

$$\text{BWD}_\varepsilon(P_\beta, P_{\text{rand}}) = \max_{g, f} \left\{ \mathbb{E}_{(s,a) \sim P_\beta}[g(s, a)] + \mathbb{E}_{(s,a') \sim P_{\text{rand}}}[f(s, a')] - \mathcal{R}_\varepsilon(g, f) \right\}, \quad (15)$$

where $g$ and $f$ are learnable potential functions, $\varepsilon > 0$ is the regularization parameter, and $\mathcal{R}_\varepsilon(g, f)$ is the entropic regularization term. With the cost $c$ from Equation equation 14, this term is:

$$\mathcal{R}_\varepsilon(g, f) = \varepsilon \mathbb{E}_{(s,a) \sim P_\beta, (s,a') \sim P_{\text{rand}}} \left[ \exp\left( \frac{g(s, a) + f(s, a') - c((s, a), (s, a'))}{\varepsilon} \right) \right]. \quad (16)$$

The potentials $g, f$ are typically parameterized by neural networks and optimized using stochastic gradient methods (Korotin et al., 2022b). It can be computed subsequent to estimating $Q_\theta^\beta$ from the static dataset, obviating the need for complete training of a separate RL agent for evaluation.

## 5 DISTANCE ANALYSIS EXPERIMENTS

This section presents empirical investigations into the efficacy of various distance metrics, including the proposed Bellman-Wasserstein Distance (BWD), for assessing offline dataset quality. We evaluate these metrics on standard benchmark environments and datasets.

**Datasets.** We utilize the Datasets for Deep Data-Driven Reinforcement Learning (D4RL) benchmark suite (Fu et al., 2020), a comprehensive collection of datasets designed for evaluating offline RL agents. D4RL encompasses a variety of tasks. For our experiments, we focus on the `Mujoco` suite (comprising Walker2d, Hopper, and HalfCheetah environments). We used the standard random, medium, medium-replay, medium-expert, expert data division.

Following the data collection principles of the D4RL, we have constructed extended existing offline datasets with a higher-devision frequency. For each environment, there are now seven quality levels to support further experimentation. All the datasets were generated by training a policy online, applying early stopping, and then collecting 1M samples from this partially trained policy. The twin delayed DDPG (TD3) Fujimoto et al. (2018) algorithm was used to train the online policy.

To eliminate the influence of random factors, the datasets were collected using a combination of data from five runs of online training with five different seeds, with each seed contributing 200,000 samples.

**Settings.** It is crucial to recognize that raw data quality or environment complexity alone does not fully determine data quality. A more indicative measure is the *oracle score*, which reflects the performance achievable by a range of strong RL algorithms on a given task. Following the evaluation protocol suggested by Furuta et al. (2021), we compute an oracle score for each task by averaging the performance of several established offline RL algorithms. Specifically, for the `Mujoco` tasks, these scores are derived from the performance of Behavior Cloning, Advantage-Weighted Actor-Critic (AWAC), OnestepRL (Brandfonbrener et al., 2021), TD3+BC (Fujimoto & Gu, 2021), Conservative Q-Learning (CQL) (Kumar et al., 2020), Implicit Q-Learning (IQL) (Kostrikov et al., 2021), and Extreme Q-Learning (XQL) (Garg et al., 2023).

To numerically evaluate data quality, we employ the methods detailed in Section 3 (Q-function and Advantage-based estimation) and our proposed Bellman-Wasserstein Distance from Section 4, alongside the PD concept from Section 3. For the PD we set $\pi = \pi_{\text{rand}}$, and evaluate $J(\pi_{\text{rand}}) - J(\beta)$.

To obtain the behavioral Q-function, $Q^\beta$, a feed-forward neural network with a single hidden layer of 256 units was trained for 10,000 steps with a batch size of 256. The input to this network consisted of the concatenated state and action. Post-training, we randomly sampled 20,000 transitions (averaged over three random seeds) to compute the scores for each metric.

To compute approximations of the Bellman-Wasserstein Distance, we employ neural networks with a single hidden layer of 256 units $g_\psi : \mathbb{R}^D \to \mathbb{R}$ and $f_\phi : \mathbb{R}^D \to \mathbb{R}$ to parameterize the potentials $g$ and $f$ in the dual formulation equation 16.

Table 1: Raw values of the metrics for rewards, critic value, and Bellman-Wasserstein distance. (Group by problem.) As dataset quality improves, we can see that BWD grows proportionally with the oracle score.

| | BASELINE | ORACLE | $A^\beta(s,a)$ | PD | $Q^\beta(s,a)$ | $r(s,a)$ | BWD |
|---|---|---|---|---|---|---|---|
| HALFCHEETAH | RANDOM | 15.26* | 0.019 | -0.125 | -52.205 | -0.286 | 0.006 |
| | MEDIUM | 44.78 | -0.095 | -13.347 | 414.775 | 4.762 | 468.918 |
| | MEDIUM-REPLAY | 41.17 | -0.538 | -33.724 | 183.582 | 3.091 | 295.577 |
| | MEDIUM-EXPERT | 80.01 | -0.436 | -17.851 | 678.412 | 7.727 | 1130.780 |
| | EXPERT | 80.40 | 0.122 | -31.876 | 1029.175 | 10.639 | 1185.976 |
| HOPPER | RANDOM | 10.55* | -0.147 | -0.272 | 10.392 | 0.836 | 21.677 |
| | MEDIUM | 62.23 | 0.020 | -1.805 | 238.403 | 3.100 | 306.393 |
| | MEDIUM-REPLAY | 75.87 | 0.037 | -5.110 | 129.268 | 2.378 | 237.555 |
| | MEDIUM-EXPERT | 92.79 | 0.295 | -2.269 | 276.679 | 3.355 | 356.766 |
| | EXPERT | 99.83 | 0.093 | -1.573 | 348.335 | 3.610 | 391.350 |
| WALKER | RANDOM | 2.03* | -0.030 | -0.158 | -1.063 | 0.090 | 0.251 |
| | MEDIUM | 76.67 | -0.311 | -16.264 | 279.726 | 3.392 | 358.277 |
| | MEDIUM-REPLAY | 60.98 | 0.004 | -18.606 | 134.724 | 2.476 | 268.269 |
| | MEDIUM-EXPERT | 103.27 | -0.552 | -14.659 | 362.572 | 4.156 | 510.431 |
| | EXPERT | 107.03 | -0.268 | -7.108 | 484.168 | 4.913 | 543.489 |

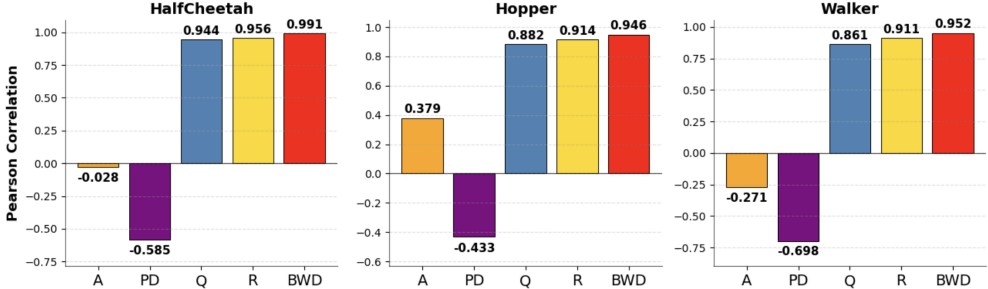

Figure 2: Pearson correlation between different metrics and the oracle score. The scores are reported over the self-crafted datasets from the different training steps and three random seeds.

**Results**: The experimental results of this analysis are presented in Table 1 and Figure 2. As can be seen, the proposed BWD algorithm has a higher correlation with the oracle score. Traditional methods provide scalar assessments that may collapse multimodal value distributions into scalars and lose information about behavioral coherence. It has been demonstrated that the proposed metrics, particularly BWD, can effectively estimate dataset quality using only a small fraction of the dataset. This is favorable compared to methods requiring full policy optimization, which typically demand access to the entire dataset and significantly more computation. These experiments across a range of D4RL datasets provide initial validation for the utility of optimal transport-based distances in characterizing offline RL data before extensive policy optimization is committed to.

**Run time.** The code is implemented in the `PyTorch` framework and will be publicly available along with the trained networks. Our method converges within 10 minutes on an Nvidia 1080 (12 GB) GPU, using only 10k updates. While our method uses additional neural networks as potentials, its clock time is $\sim$**10x faster** than the alternative of training a full policy to convergence to understand dataset quality.

## 6 POLICY REGULARIZATION EXPERIMENTS

This section describes experiments designed to evaluate the effectiveness of using the proposed Bellman-Wasserstein distance as a regularization term during policy optimization.

The primary objective of these experiments is to assess whether explicitly maximizing the dissimilarity to a random policy, as quantified by BWD, can enhance policy performance. The underlying assumption is that a random policy typically executes suboptimal or irrelevant actions; thus, guiding

the learned policy away from such random behaviors should be beneficial. We integrate the BWD-based regularization terms with one of contemporary offline RL algorithms: Implicit Q-Learning (IQL) Kostrikov et al. (2021). For experiments involving IQL, we utilized the CORL library Tarasov et al. (2022).

To simulate the random policy, actions were sampled from a normal distribution, as described in Section 4. During policy optimization, the critic function $Q_\theta^\beta : \mathbb{R}^{|S|} \times \mathbb{R}^{|A|} \to \mathbb{R}$, approximated by a neural network, is trained on the provided expert data using an update rule such as Eq 5. The $Q^\beta$ values were computed using standard critic architecture for IQL. The regularized objective can be expressed as:

$$J_{\text{reg}}(\pi_) = J_{\text{original}}(\pi_) + \text{BWD}(\pi_, \pi_{\text{rand}}).\tag{17}$$

The hyperparameters for the base IQL algorithms were kept consistent with their standard configurations for the respective tasks. For the potential networks $g_\psi$ and $f_\phi$ used in BWD computation, single hidden-layer neural networks with 256 units were utilized. The regularized algorithm were trained for $10^6$ updates with a batch size of 256. No pre-trained models were used in these experiments.

**Results**: The experimental results, presented in Figure 3 and Table 2, demonstrate the impact of BWD regularization on policy performance across three locomotion tasks. For HalfCheetah, the normalized scores show consistent improvement with lower dataset quality. The Hopper task exhibited more variability in performance, suggesting greater sensitivity to the regularization approach. Walker results showing the most superior performance, indicating that BWD regularization benefits from additional state information. These findings support our hypothesis that explicit dissimilarity maximization to random policies can enhance policy optimization in offline RL settings.

Table 2: The average normalized scores of the default IQL and IQL incorporating the BWD distance, computed over three random seeds.

| | METHOD | 50K | 200K | 400K | 600K | 800K | 1000K |
|---|---|---|---|---|---|---|---|
| HALFCHEETAH | IQL | 32.943 | 57.187 | 69.730 | 75.424 | 72.319 | 82.159 |
| | IQL+BWD | 32.570 | 56.691 | 70.756 | 76.328 | 75.986 | 86.525 |
| HOPPER | IQL | 9.711 | 40.975 | 80.209 | 83.395 | 91.693 | 95.031 |
| | IQL+BWD | 10.990 | 46.192 | 88.884 | 84.447 | 102.376 | 89.067 |
| WALKER | IQL | 8.108 | 26.130 | 71.307 | 76.723 | 78.232 | 81.149 |
| | IQL+BWD | 7.212 | 28.898 | 72.649 | 64.944 | 89.408 | 98.545 |

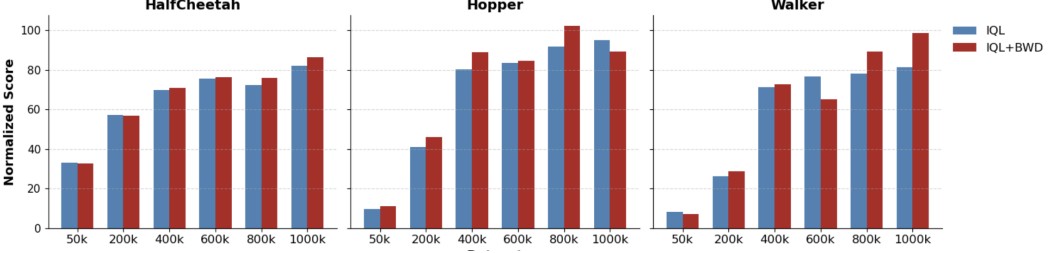

Figure 3: Results of including BWD distance into the training process of IQL. The average normalized scores are reported over the datasets from the different training steps. The performance results are averaged over the three random seeds.

**Run time.** The code is implemented in the `PyTorch` frameworks and will be publicly available along with the trained networks. Our method converges within 2–3 hours on Nvidia 1080 (12 GB) GPU. We used WandB for babysitting training process. The code is available in supplementary materials.

# 7 RELATED WORK

Considering from the distributional perspective we can say that our method related to behavior cloning methods like Primal Wasserstein Imitation Learning Dadashi et al. (2020) as well as Sinkhorn

Imitation Learning Papagiannis & Li (2020) used optimal transport between to minimize the occupancy distributions to the imitator. Fickinger et al. (2021) extended the Wasserstein distance to the cross-domain setting for the agents that live in different spaces. In the cross-domain imitation learning problem, Wasserstein distance-based methods have been developed to measure the discrepancy between policies in different domains Fickinger et al. (2021).

All referenced OT based distances method in offline RL and behaviour cloning aim at matching the state-action distribution of the learner with the distribution of the expert. This allows to move agent distribution of actions closer to the expert policy.

Recently, several methods have been proposed to estimate the complexity of RL tasks. Tabular MDP settings are the focus of most of these methods. For example, Jiang et al. (2017) proposed *Bellman rank* and showed that MDPs with a low Bellman rank can be solved more efficiently.

For the metric that can be approximated numerically for complex RL tasks, Oller et al. (2020) proposed a method called random weight guessing. This method runs a set of policies with random weights without any training, and it was shown that the mean, percentile, and variance of the episodic rewards are related to the difficulty of finding a good policy by random search.

Lately, a method called *policy information capacity* has been proposed (Furuta et al., 2021), the idea is to measure the mutual information between policy parameters and cumulative episode rewards. Compared to these methods, in this paper we consider task complexity estimation for offline RL.

## 8  CONCLUSION

This work introduced the Bellman-Wasserstein distance, a novel metric to assess the quality of the offline RL dataset by quantifying the deviation of the behavioral policy from random actions. Our experiments demonstrated that incorporating BWD as a regularization term during policy optimization improves performance across locomotion tasks, particularly in datasets with mixed-quality trajectories. These results validate that (1) dissimilarity to random policies correlates with data quality, and (2) distributional metrics like BWD can guide offline RL training without costly agent pretraining. Future work could explore BWD's applicability to broader algorithm classes and its role in automated dataset curation. As a **limitation**, we can consider the fact that to apply our method, one has to train potential function $g$ and $f$ which may be non-trivial.

**Future work.** General optimal transport problem allows us to use arbitrary cost functions and enhance future work in the direction of finding better RL specific cost functions. The general OT duality formula *sums up* previously known formulas for *weak* and *strong* functionals Korotin et al. (2022a), (Korotin et al., 2021b, Eq. 9), (Rout et al., 2022, Eq. 14), (Fan et al., 2021, Eq. 11), (Henry-Labordere, 2019, Eq. 11), (Gazdieva et al., 2022, Eq. 10). We hope that our method will encourage RL towards more general, scalable, and stable deep offline algorithms.

**Societal Impact.** This paper presents research aimed at advancing the field of RL. Although our work may have various societal implications, we do not believe that any particular ones need to be specifically emphasized here.

**Reproducibility.** To reproduce our experiments, please refer to the supplementary materials. The code will be open-sourced. Details on the hyperparameters used are presented in the settings.

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

# A APPENDIX

---

**Algorithm 1** Estimating the Bellman–Wasserstein Distance (BWD)

---

**Require:** Offline dataset $D = \{(s, a, r, s')\}$; discount $\gamma$; random policy $\pi_{\text{rand}}(\cdot|s)$; entropic weight $\varepsilon$; critic steps $T_Q$; OT steps $T_{\text{OT}}$; batch size $B$; negatives per state $K$

1: **Fit behavioral critic** $Q_\beta$ on $D$ (*SARSA-style*; Eq. (5)):

  For $t = 1..T_Q$: sample $(s, a, r, s') \sim D$, draw $a^+ \sim D(\cdot|s')$, minimize $\big(r + \gamma\, Q_\beta(s', a^+) - Q_\beta(s, a)\big)^2$ w.r.t. $Q_\beta$.

2: Build empirical marginals: $P_\beta$ over $(s, a)$ from $D$; $P_{\text{rand}}$ over $(s, a')$ with $s \sim D_S$ and $a' \sim \pi_{\text{rand}}(\cdot|s)$ (Sec. 4).

3: Initialize potentials $g_\psi(s, a)$ and $f_\phi(s, a')$.

4: **for** $t = 1..T_{\text{OT}}$ **do**

5:   Sample $\{(s_i, a_i)\}_{i=1}^B \sim P_\beta$; for each $i$, sample $\{a'_{i,k}\}_{k=1}^K \sim \pi_{\text{rand}}(\cdot|s_i)$.

6:   **State-conditional costs:** $c_{i,k} = Q_\beta(s_i, a'_{i,k}) - \|a'_{i,k} - a_i\|_2^2$        (Eq. 14)

7:   **Dual objective (entropic OT):**

$$\mathcal{L} = \frac{1}{B}\sum_{i=1}^B g_\psi(s_i, a_i) + \frac{1}{BK}\sum_{i,k} f_\phi(s_i, a'_{i,k}) -$$

$$\varepsilon\,\frac{1}{BK}\sum_{i,k}\exp\Big(\frac{g_\psi(s_i, a_i) + f_\phi(s_i, a'_{i,k}) - c_{i,k}}{\varepsilon}\Big) \qquad\qquad \text{(Eqs. 15- 16)}$$

8:   Update $\psi, \phi$ by *gradient ascent* on $\mathcal{L}$.

9: **end for**

10: **Return** $\widehat{\text{BWD}}_\varepsilon(P_\beta, P_{\text{rand}})$ by evaluating the objective on a held-out minibatch. Larger values $\Rightarrow$ higher dataset quality (Table 1, Fig. 2).

---

**Algorithm 2** IQL with BWD Regularization (high level)

---

**Require:** Dataset $D$; weight $\lambda_{\text{BWD}}$; random policy $\pi_{\text{rand}}$; $\varepsilon$

1: Initialize IQL (actor $\pi_\theta$, critic/value) and BWD potentials $(g_\psi, f_\phi)$.

2: **for** training step $t = 1..T$ **do**

3:   Update IQL critic/value on a minibatch from $D$ (standard IQL).

4:   Sample states $\{s_i\}_{i=1}^B \sim D_S$, set $a_i = \pi_\theta(s_i)$, and sample $\{a'_{i,k}\} \sim \pi_{\text{rand}}(\cdot|s_i)$.

5:   Compute a batch estimate $\widehat{\text{BWD}}_\varepsilon$ using Eqs. 15- 16 with fixed $Q_\beta$; *stop gradient* through $(g_\psi, f_\phi)$ for the actor step.

6:   Actor update: ascend $\nabla_\theta\big(J_{\text{IQL}}(\theta) + \lambda_{\text{BWD}}\widehat{\text{BWD}}_\varepsilon\big)$; optionally take a few ascent steps on $(g_\psi, f_\phi)$.

7: **end for**

---

