# OpenReview forum: "Expert or not? Assessing data quality in offline reinforcement learning"
_ICLR.cc/2026/Conference — ICLR 2026 Conference Withdrawn Submission_

### Official Review · Reviewer_sUwW · 2025-10-16

**Soundness:** 2
**Presentation:** 3
**Contribution:** 2
**Rating:** 2
**Confidence:** 4

**Summary:**

This paper introduces a new metric, the Bellman-Wasserstein Distance (BWD), intended to assess the quality of offline RL datasets without requiring full agent training. The BWD uses optimal transport to measure a value-aware dissimilarity between the dataset's behavioral policy and a random policy, with the core idea that high quality data should be very distinct from random behavior. This metric is tested in two ways: as a predictor of offline RL performance on a few continuous control D4RL tasks, and as a regularizer to improve policy learning of the IQL algorithm.

**Strengths:**

1. The paper tackles a well motivated and practical problem. An efficient, reliable method for a priori dataset quality assessment would be a valuable tool for the offline RL community.
2. The idea of using a value-aware distributional distance to quantify deviation from a random baseline seems novel.
3. The "oracle score" used as ground truth for data quality is methodologically sound.

**Weaknesses:**

**BWD cost function:**

The paper's central contribution, the BWD cost function, is fundamentally flawed. Fig 2 results show that high BWD correlates with high data quality. However, the cost function (c) is maximized when random actions a′ have high Q values and behavioral actions (a) are very similar to them, in contradiction to what is stated in algorithm 1, and to the results in Fig 2. If I understand correctly, the BWD metric should indeed be highest for datasets that most resemble random behavior, which is the exact opposite of the paper's premise.

**Confused definition of "data quality"**:

The paper frames “data quality” as "expert-ness", but the ground-truth oracle score is the aggregated performance of algorithms (like IQL) designed to stitch suboptimal trajectories (also stated in lines 47-48 “offline RL algorithms are more efficient with suboptimal data”). This implies that the definition of "data quality" is simply a dataset that makes offline RL algorithms succeed.

**Marginal gain:**

The BWD calculation requires a pre-trained behavioral critic Q_beta. The presented results (Fig. 2) show only a marginal improvement in correlation for using BWD over simply using the average Q_beta value (e.g., 0.991 vs 0.956 for HalfCheetah). The significant added complexity of learning two additional potential functions for the OT problem is not justified by this minor gain.

**Regularization results:**

The claim that BWD serves as a useful regularizer is not convincingly supported. On the Hopper task, the performance of IQL+BWD results significantly below IQL. While the authors consider this as "variability," it suggests the regularizer is not stable enough. In order to be considered as stable regularizer, the BWD should be combined with the other baselines as well.

**Limited experimental validation:**

The validation is confined to three D4RL MuJoCo locomotion tasks. These dense-reward environments are a "safe" choice where learning a behavioral critic Q​ is feasible. The paper should test its metric on more challenging real-world dagtasets, or datasets with sparse-reward tasks (e.g., AntMaze) where Q is difficult to estimate and stitching is paramount.

**Questions:**

Please address the weaknesses above.

---

### Official Review · Reviewer_eLaN · 2025-10-20

**Soundness:** 2
**Presentation:** 2
**Contribution:** 2
**Rating:** 2
**Confidence:** 4

**Summary:**

The paper introduces a novel metric, called the Bellman–Wasserstein distance (BWD), to measure the quality of an offline dataset. The idea behind this metric is to leverage optimal transport to compute the dissimilarity between the behavioral policy and a reference policy (random policy). This metric does not require training an actor, but simply a critic, leading to a much more efficient training process. The authors show that BWD can be used as an additional regularization term for an offline reinforcement learning algorithm.

**Strengths:**

The paper is easy to follow and the idea is interesting

**Weaknesses:**

- There is a missing comparison with existing approaches and recent literature from the past two years.
- The experiments are not very convincing, and the statistical significance is unclear (confidence intervals are missing).

**Questions:**

- The problem of understanding data quality in offline reinforcement learning (RL) and designing adaptive offline methods is an old problem in RL, with a recent surge of interest from the community. While I found the approach interesting, the authors have not considered or compared several approaches that have been published in the last two years. For example,

Anikait Singh, Aviral Kumar, Quan Vuong, Yevgen Chebotar, Sergey Levine:
Offline RL With Realistic Datasets: Heteroskedasticity and Support Constraints. CoRR abs/2211.01052 (2022)

Zhang-Wei Hong, Pulkit Agrawal, Remi Tachet des Combes, Romain Laroche:
Harnessing Mixed Offline Reinforcement Learning Datasets via Trajectory Weighting. ICLR 2023

Zhang-Wei Hong, Aviral Kumar, Sathwik Karnik, Abhishek Bhandwaldar, Akash Srivastava, Joni Pajarinen, Romain Laroche, Abhishek Gupta, Pulkit Agrawal:
Beyond Uniform Sampling: Offline Reinforcement Learning with Imbalanced Datasets. NeurIPS 2023

Tenglong Liu, Yang Li, Yixing Lan, Hao Gao, Wei Pan, Xin Xu:
Adaptive Advantage-Guided Policy Regularization for Offline Reinforcement Learning. ICML 2024

Edoardo Cetin, Andrea Tirinzoni, Matteo Pirotta, Alessandro Lazaric, Yann Ollivier, Ahmed Touati:
Simple Ingredients for Offline Reinforcement Learning. ICML 2024

Yunpeng Qing, Shunyu Liu, Jingyuan Cong, Kaixuan Chen, Yihe Zhou, Mingli Song:
A2PO: Towards Effective Offline Reinforcement Learning from an Advantage-aware Perspective. NeurIPS 2024

Yixiu Mao, Qi Wang, Chen Chen, Yun Qu, Xiangyang Ji:
Offline Reinforcement Learning with OOD State Correction and OOD Action Suppression. NeurIPS 2024

Yiqin Yang, Quanwei Wang, Chenghao Li, Hao Hu, Chengjie Wu, Yuhua Jiang, Dianyu Zhong, Ziyou Zhang, Qianchuan Zhao, Chongjie Zhang, Bo Xu:
Fewer May Be Better: Enhancing Offline Reinforcement Learning with Reduced Dataset. ICLR 2025


- As a metric, BWD appears to be only marginally better than the reward of the behavioral Q-function. Why should we use this metric? As a regularization term, it improves over IQL, but we do not know how its performance compares to more recent offline RL algorithms.
- Are the results statistically significant? You did not report confidence intervals for your experiments.


Overall I think the paper needs a major revision to account for these points.

---

### Official Review · Reviewer_ywkk · 2025-10-21

**Soundness:** 2
**Presentation:** 2
**Contribution:** 2
**Rating:** 2
**Confidence:** 4

**Summary:**

The paper proposes a new metric to assess the quality of a transition dataset to predict (in terms of correlation) the performance offline RL algorithms. The proposed metric is cheaper to compute than actually training offline RL algorithms, thus making it a viable approach for the problem. The authors empirically investigate the correlation between the proposed metric and the actual performance and they also propose a way to integrate it as a regularization into offline RL algorithms themselves.

**Strengths:**

* The problem studied in the paper is very important, since the performance of offline RL algorithms is significantly affected by the type of data available and a proper way to assess the correlation between the nature of the data and the performance is lacking.
* The authors propose a number of different metric and provide preliminary evidence of the stronger correlation between the BWD metric and the average performance of offline RL algorithms.

**Weaknesses:**

* The actual focus of the paper is somehow unclear to me. From the introduction, my impression is that the objective is to determine the quality/nature of the data, since this does impact the performance of different approaches differently. For instance, if the data contains several expert trajectories, a BC-first approach is preferable, whereas when data become more transition-based with good coverage, it is better to use "pure" offline RL. Finally, if data are transition-based and poor coverage, then they are just "poor" and they do not support any good learning performance. I think this perspective is well captured in Fig.1. In this sense, I would have expected the paper to focus more on a model/algorithm selection objective: the metric evaluating the nature of the data could be used to decide which algorithm or which parameter is best for this specific dataset, without the need of actually running all possible algorithms. On the other hand, in the end the authors evaluate the correlation between the data and the *average* performance of a pool of different offline RL algorithms. This seems more about assessing a sort of average "quality" of the data but it doesn't seem to inform any algorithmic decision but rather to "rank" different datasets. While this perspective could still be valuable in terms of choosing which dataset to use, I'm not sure it would be very effective. Consider the following case: Dataset D1 has an average quality Q1 > Q2, which is the average quality of a dataset D2. This doesn't mean that the best offline RL algorithm would not be able to obtain a very strong performance using D2. So overall, I'm not convinced about the motivation and the actual significance of the overall contribution.
* On a technical side, there are multiple aspects of the current paper that are not clear and would need more intuition (see questions below). Also, the authors do not try to make any connection with the vast theoretical literature on offline RL, where notions such as coverage, concentrability coefficients, and so on and widely studied as crucial conditions influencing the performance of offline RL (see e.g., https://arxiv.org/pdf/2508.07746v1)
* On the empirical side, there is partial evidence the BWD is better correlated to the average performance than other metrics, but the gap does not seem major. Similarly, when BWD is used a regularization for IQL, the improvement is not very consistent and it is not clear if it is statistically significant or not.

**Questions:**

* The idea behind BWD is not fully clear to be. The term Q^beta(s,a') is measuring the one step improvement (or more likely worsening) that a random policy would have wrt to the beta policy and this is used to weight the transport cost of moving from beta to random policy. A few questions
** Q^beta(s,a') is unlikely to be well estimated for a' actions that are not well supported by beta and the data. This is pretty much the same problem we have for offline RL, so if BWD can be well estimated, then I suspect offline RL could also work, while if data have poor coverage, then BWD would not be estimated accurately.
** The second term ||a'-a|| is independent on the state and it would be estimating how much the action a chosen by beta differs from the average action chosen by pi_rand. Why is this a relevant cost?
** When BWD is used in offline RL, it is written as BWD(pi, pi_rand), which suggests that now it evaluates how worse pi_rand would be compared to pi, which now would not be really connected to the data and the behavior policy and it would be even harder to accurately evaluate from data itself.
* Eq. 5 should be pi->beta?
* Eq.10 is basically J(\beta), since beta is assumed to be the policy generating the data.
* Eq.12 should be mostly close to 0, since a~D and a~beta should be pretty much the same distribution?
* Fig2: it seems like here you use a different set of datasets then in Tab.1, why?
* Tab.1 is not very effective to read, a plot would make it easier to assess the correlation.
* Tab2 and Fig3 do not report confidence intervals. Given the narrow margin of some differences, it is hard to assess their statistical significance.
* In defining pi_rand for the regularized algorithm, you use a normal distribution. This is not covering uniformly the action space, so I'm wondering how the choice of the mean could bias the algorithm.

Minor
* The paper is not very well written and many sentences do not real well, eg, L154-156,
* L220: "optimal" -> "exact"?

---

### Official Review · Reviewer_ApvF · 2025-10-31

**Soundness:** 1
**Presentation:** 2
**Contribution:** 2
**Rating:** 2
**Confidence:** 5

**Summary:**

The paper studies the problem of assessing the quality of an Offline RL dataset by means of the Bellman-Wasserstein Distance (BWD).
It further investigates using the BWD as a regularizer during training policies in the offline RL setting.
The paper gives an introduction to (offline) RL and optimal transport, followed by a review of some prior methods for assessing the quality of data in the RL setting.
Based on this, the BWD is introduced,  as the dual form of an optimal transport problem.
The experimental evaluation considers (1) the similarity between BWD and an oracle score and (2) using BWD as a regularizer on top of IQL.

**Strengths:**

- The idea of defining (RL) data quality as distance to random behavior is intuitive and actionable, which focuses deliberately on the actions and is independent of the visited states.
- In offline RL, many algorithms aim for conservative learning that either penalizes unseen actions or incentivices actions seen in the dataset. Having an explicit regularizer to not act randomly is an interesting idea to aim for less pessimistic behavior.

**Weaknesses:**

- The paper does not discuss the very related prior work Schweighofer et al. 2022, which study the effect of dataset quality on the performance of Offline RL algorithms. Particularly, their introduced measures TQ and SACo can be calculated without any learning involved, they are essentially summary statistics that capture the quality of RL datasets. Comparing the usefulness of BWD over those measures would strengthen the first contribution of the paper.
- While the second part of the empirical evaluation, using BWD as a regularizer, makes a lot of sense to me, I found the first part of the empirical evaluation not very convincing. It is still not clear to me what the benefit of having a measure of dataset quality should be. In the referenced Schweighofer et al. 2022 it is used to analyze the dependence of algorithms on the mix. In the present work, I do not see any immediate implications from the presented finding. Example: Suppose I have a dataset and I measure its quality using BWD, the result is 2357, what do I now do with this information?
- I do not understand the added benefit of BWD over Q and R (Table 1 and Figure 2) in the first set of experiments. They all give the same ranking of considered datasets in Table 1 and as discussed in the question below I think that pearson correlation is not justified. While Q might be somewhat hard to calculate, R is just an average over rewards observed in the dataset, which is trivial to compute. However, I would argue that trajectory returns (see the TQ measure in Schweighofer et al. 2022) is even better as it circumvents some of the problems the authors mentioned for the cumulative reward. Also, the notion of SACo could explain why the ranking of oracle is different than for BWD, R and Q on Hopper medium vs. medium-replay.
- Neither Table 2 not Figure 3 feature the deviations over the runs, only showing average performance, making it impossible to assess the significance of the improvement.
- Neither the dataset used for training nor the normalization parameters for the results in Table 2 / Figure 3 are discussed, making it hard to assess them.
- Only a single offline RL algorithms (IQL) is used as a baseline in the Experiments in section 6. Having at least 3 algorithms with and without the additional BWD penalty would show the usefulness much more convincingly. For example, CQL, TD3+BC or even just on top of BC would be super interesting.

Some minor weaknesses:
- I do not think that Eq.(5) is correct. The inner expectation should also be on the outside, i.e. $E_{(s,a,r,s',a')\sim D}$ which is why it is called a SARSA-style objective. Implementing the sampling the way it is currently written might be tricky and is not the way this objective was considered in prior work such as the referenced Kostrikov et al. (2021).
- I am worried about the way the value function is apparently calculated (line 237). There is no guarantees that the Q-value function is a good approximation anywhere else than for tuples (s,a) within the dataset. However, the value function is evaluated as an expectation over actions that have never been observed for this state. The inability to estimate Q(s,a) correctly for unseen state-action pairs is the primary issue that Offline RL is tackling and I feel this is swept under the rug here. At least, this issue needs to be mentioned, even if the advantage function is not fundamental to the contribution of this work.

Grammar and presentation should be improved for the next version of this work.
Some concrete points:
- Abbreviations are introduced multiple times (e.g. D4RL and TD3), others are never introduced (IQL)
- Use \eqref, \citet and \citep in a consistent and correct way. I know this is a bit picky, but it is very easy to implement and makes the paper much more appealing visually.
- Missing year in citation in line 147.
- Line 153 is not a proper sentence.
- Line 124: "... we **estimate** the parameters ..."
- Line 171: "... c(x, y) measure**s** how ..."
- Line 189 has word "methods" twice one after another. Also, I would spend less space introducing optimal transport and write something about the neural network based methods used for this work, which is much more important.
- Most of the python packages cited for runtime (e.g. PyTorch) has a publication one could cite.

Schweighofer, Radler, ..., Hochreiter (2022) A Dataset Perspective on Offine Reinforcement Learning, CoLLAs

**Questions:**

- Why does using the pearson correlation in Figure 2 makes sense? Are there any arguments why the assumptions of pearson correlation are justified? If not, (e.g. spearman) rank correlation might be the more suitable method of investigation.
- How are the distribution parameters (mean and variance) for simulating the random policy (line 382) chosen? The reader is defered to section 4 for details, but I could not find them there. To me it is not obvious how one would set that, especially under a Gaussian assumption, a uniform distribution over "safe" actions (torques) would make more sense for me in D4RL environments.

---

### Note · Authors · 2025-12-03

I have read and agree with the venue's withdrawal policy on behalf of myself and my co-authors.